# Investigation of Chemical Composition and Biological Activities of *Ajuga pyramidalis*—Isolation of Iridoids and Phenylethanoid Glycosides

**DOI:** 10.3390/metabo13010128

**Published:** 2023-01-14

**Authors:** Anthonin Gori, Benjamin Boucherle, Aurélien Rey, Maxime Rome, Caroline Barette, Emmanuelle Soleilhac, Christian Philouze, Marie-Odile Fauvarque, Nicola Fuzzati, Marine Peuchmaur

**Affiliations:** 1Univ. Grenoble Alpes, CNRS, DPM, 38000 Grenoble, France; 2CHANEL Parfums Beauté, 93500 Pantin, France; 3Univ. Grenoble Alpes, CNRS, Jardin du Lautaret, 38000 Grenoble, France; 4Univ. Grenoble Alpes, CEA, Inserm, IRIG, BGE, 38000 Grenoble, France; 5Univ. Grenoble Alpes, CNRS, DCM, 38000 Grenoble, France

**Keywords:** *Ajuga pyramidalis*, chemical composition, iridoids, phenylethanoid glycosides, antioxidant, epidermal renewal

## Abstract

Despite several studies on the *Ajuga* L. genus, the chemical composition of *Ajuga pyramidalis*, an alpine endemic species, is still largely unknown. The purpose of this study was to therefore deeper describe it, particularly from the phytochemistry and bioactivity perspectives. In that respect, *A. pyramidalis* was investigated and 95% of the extracted mass of the plant was characterized by chromatography and mass spectrometry. Apart from the already determined chemical compounds, namely, harpagide and 8-*O*-acetylharpagide, two iridoids, and neoajugapyrin A, a neo-clerodane diterpene, and three polyphenols (echinacoside, verbascoside and teupoloside) were identified for the first time in *A. pyramidalis*. Incidentally, the first RX structure of a harpagoside derivative is also described in this paper. The extracts and isolated compounds were then evaluated for various biochemical or biological activities; notably a targeted action on the renewal of the epidermis was highlighted with potential applications in the cosmetic field for anti-aging.

## 1. Introduction

The plant family *Lamiaceae* comprises more than 7000 species and approximately 236 genera, spread worldwide [1]. Many studies have highlighted that herbs from this family show potent antioxidant and antibacterial activities, mostly due to the quantity and quality of the phenolic compounds present in them [2]. The *Lamiaceae* family is divided into 12 subfamilies [3,4] among which is the subfamily *Ajugoideae*, which contains 26 genera and ca. 760 species, distributed worldwide, but which is primarily found in tropical regions [5]. The genus *Ajuga* L., belonging to this subfamily, comprises approximately 90 species, mostly distributed in the north temperate zones of Europe, Africa and Asia [6]. Many of these plants have been studied, either for medicinal uses, for pharmacological activities or for phytochemical studies [7,8,9,10,11]. In particular, some *Ajuga* species have been traditionally used in ethnomedicinal practices for anti-inflammatory activity, expectorant effect, protective action against heart diseases, stomach pain or for the treatment of jaundice, asthma or even cancer.

*Ajuga pyramidalis*, the pyramidal bugle, is distributed in all Europe, with a clear preference for the boreal and/or mountainous areas, such as the Alps [12]. This 5 to 20 cm plant was named after its general appearance, a pyramid shape with green leaves at the base and burgundy red at the apex.

While more than 280 chemical constituents have already been isolated and identified from the genus *Ajuga* L. [7,13], to date, only two phytochemical studies have been conducted on *A. pyramidalis* [14,15]. One of them described the identification of two iridoids, harpagide and 8-*O*-acetylharpagide, while the other specified the discovery of a new *neo*-clerodane diterpene, ajugapyrin A (Figure 1), which was later found in *Scutellaria galericulata*; on that occasion, its structure was revised and renamed neoajugapyrin A [16]. In both cases, the identified compounds are known as chemotaxonomic markers for *Ajugoideae* [17]. Furthermore, these secondary metabolite families together with flavone glycosides are known to endow interesting biological activities, such as anti-inflammatory, antitumor, anti-oxidant, antimicrobial and antimutagenic effects [7,13].

In our research for broadening knowledge on alpine plants, *A. pyramidalis* was investigated and 95% of the extracted mass of the plants was characterized by chromatography and mass spectrometry compared to analytical compounds when necessary. In addition, five compounds were isolated from the secondary metabolite-enriched extract and were identified as the two iridoids previously determined in *A. pyramidalis*, and three polyphenols (echinacoside, verbascoside and teupoloside) were never described in *A. pyramidalis* but were already known in the genus *Ajuga* L. The biological activities of some extracts and isolated compounds were then investigated on purified tyrosinase as well as on different kinds of human cultured cells regarding their potential activity on key cell parameters, such as cell viability, cytokines release (IL-8), stress response (autophagy), and the expression levels of a set of genes relevant for cosmetic applications.

## 2. Materials and Methods

### 2.1. General Experimental Procedures

All solvents used for the preparation of the samples and analyses were of analytical grade and purchased from VWR SAS. All standards and chemical reagents were supplied from Sigma-Aldrich (Saint Quentin Fallavier, France) or VWR (Rosny-sous-Bois, France) and were used without further purifications. NMR spectra were recorded at room temperature in deuterated solvent (CD_3_OD or DMSO-*d*6) on a Brüker Avance-400 (400 MHz for ^1^H spectra, 101 MHz for ^13^C spectra) or on a Brüker Avance III 500 (500 MHz for ^1^H spectra, 125 MHz for ^13^C spectra). Chemical shifts (δ) are reported in parts per million (ppm) relative to the solvent [^1^H: δ(DMSO-*d*6) = 2.50 ppm, δ(CD_3_OD) = 3.31 ppm; ^13^C: δ(DMSO-d6) = 39.5 ppm, δ(CD_3_OD) = 49.0 ppm].

### 2.2. Plant Collection and Preparation

The whole plants (for intra-species study) or the flowered aerial parts of the *Ajuga* species were collected during spring 2018 (and during spring 2019 and 2020 for the intra-species study) around the Col du Lautaret region (45.0242, 6.2358) for *A. pyramidalis* and *A. genevensis*, around Grenoble (45.1955, 5.7794) for *A. reptans* L. and (45.2012, 5.7245) for *A. chamaepitys* in accordance with good practice for plant collection and the international treaties on biodiversity. The species were identified by Dr Maxime Rome, botanist at the Lautaret botanical garden. A voucher specimen of *A. pyramidalis* was deposited at the herbarium of the National Museum of Natural History, Paris (France), collection: vascular plants (P), specimen P00877367.

The fresh aerial parts were air-dried at room temperature until no weight loss was observed. Then, they were ground into powder using a Retsch Grindomix GM 200 and then stored in dark conditions.

### 2.3. Extraction Methodologies

Ethanolic extractions were conducted using Precellys 24^®®^ (Bertin Technologies) on 300 mg of plant material in 3.0 g of EtOH in an appropriate cartridge (7 mL). The extraction lasted for three cycles of 15 sec at 6000 rpm with 5 sec breaks between each cycle.

Extractions of increasing polarities were performed by three sequential maceration following an experimental procedure adapted from Gori et al. [18] using three solvent mixtures: heptane/ethyl acetate/acetonitrile/butan-1-ol (87:6:5:2) (% volume) for the apolar phase, heptane/ethyl acetate/acetonitrile/butan-1-ol/water (4:21:43:13:19) for the intermediate phase and ethyl acetate/acetonitrile/butan-1-ol/water (6:28:6:60) for the polar phase. The powdered aerial plant material was placed in a 2 L round bottom flask. The appropriate volume of the apolar solvent mixture ((10:1 (*v*/*w*)) was first added and the suspension was stirred at 35 °C during 24 h. After this time, the suspension was filtered. The plant residues were extracted again with the successive solvent mixtures of increasing polarities, namely intermediate and then polar ((20:1 (*v*/*w*)). Filtrates were evaporated and weighed to determine the yields.

### 2.4. Lipid Fraction Quantifications

Lipid fraction quantification was obtained by dissolving 0.6 g of apolar, intermediate or polar extract in methanol (2.4 g) and HCl (1 M, 0.3 g). The solution was stirred and then heated at 80 °C for 15 min. After cooling to room temperature, the solution was extracted by cyclohexane (3.0 g) and the resulting cyclohexane solution was analyzed by GC coupled to MS/FID (see method below).

### 2.5. Free Amino Acids Quantification

Free amino acids quantification was determined by HPLC (see method below) coupled with DAD and fluorescence detector after derivatization with orthophtalaldehyde and 9-fluorenylmethyl chloroformate using internal calibration by norvaline and sarcosine [19,20,21].

### 2.6. Chromatographic Conditions

#### 2.6.1. HPLC Method for Free Amino Acids Quantification

Analyses were performed with an Agilent 1200 infinity system (Agilent Technologies) using ELSD Sedex 100 and DAD (280 nm) detectors, an Agilent Zorbax Eclipse Plus C18 RRHD (2.1 × 50 mm, 1.8 μm) as the stationary phase, with a mobile phase composed of A = H_2_O and Na_2_HPO_4_ (10 mM), Na_2_B_4_O_7_ (10 mM), pH = 8.2 and B = MeOH:acetonitrile:H_2_O, 45:55:10 pH = 8.2 with a gradient from 98:2 to 0:100, over 13 min, 0.2 mL/min, column temperature: 40 °C, sample volume injection: 1 μL.

#### 2.6.2. HPLC Method for Carbohydrates Quantification

Analyses were performed with an Agilent 1200 infinity system (Agilent Technologies (Les Ulis, France)) using ELSD Sedex 100 and DAD (280 nm) detectors, an Acquity UPLC BEH amide (2.1 × 100 mm, 1.7 μm) as the stationary phase, with a mobile phase composed of acetonitrile:H_2_O, 9:1 and NH_4_OH (10 mmol/L) with isocratic elution over 21 min, 0.2 mL/min, column temperature: 50 °C, sample volume injection: 1 μL.

#### 2.6.3. GC Method for Lipids Quantification

Analyses were performed with an Agilent 5975N (Agilent Technologies) using MS/FID (Agilent MSD5975) detector, a 100% polydimethylsiloxane (Agilent VF-1MS) (0.25 µm × 0.25 mm × 60 m) as the stationary phase, with Helium as the mobile phase, over 95 min, 1 mL/min, column temperature: 250 °C, sample volume injection: 0.5 μL.

#### 2.6.4. HPLC Method for Organic Acids Quantification

Analyses were performed with an Agilent 1200 infinity system (Agilent Technologies) using ELSD Sedex 100 and DAD (280 nm) detectors, an Acquity UPLC BEH amide (2.1 × 100 mm, 1.7 μm) as the stationary phase, with a mobile phase composed of A = acetonitrile:H_2_O, 9:1 and NH_4_OH (10 mmol/L) and B = acetonitrile:H_2_O, 5:5 and NH_4_OH (10 mmol/L) with a gradient from 98:2 to 2:98, over 15 min, 0.5 mL/min, column temperature: 50 °C, sample volume injection: 1 μL.

#### 2.6.5. HPLC Method for Iridoids, Phenolic and Terpenic Derivatives Quantification and for Inter- and Intra-Species Studies

Analyses were performed with an Agilent 1290 infinity system (Agilent Technologies) using ELSD Sedex 100 and DAD (280 nm) detectors, an Agilent Zorbax Eclipse Plus C18 RRHD (2.1 × 50 mm, 1.8 μm) as the stationary phase, with a mobile phase composed of A = H_2_O and 0.1% formic acid and B = MeOH with a gradient from 95:5 to 5:95, over 17 min, 0.55 mL/min, column temperature: 50 °C, sample volume injection: 1 μL.

### 2.7. Derivatization of Harpagoside and Cristallographic Experiments

#### 2.7.1. Derivatization of Harpagoside

To a solution of commercial harpagoside (50 mg) in 1 mL of pyridine, eight equivalents of acetic anhydride was added dropwise at room temperature. After stirring for 24 h, the reaction mixture was heated for 4 h at 70 °C, then it was stirred for 168 h at room temperature. After concentration under reduced pressure, the residue was dissolved in CH_2_Cl_2_ and successively washed with water, a 10% aqueous solution of HCl, a saturated aqueous solution of NaHCO_3_ and finally brine. The organic phase was dried over Na_2_SO_4_ and concentrated under vacuum. The unpurified major product, a pentaacetylated harpagoside, was used for the crystallization assay without further purification.

#### 2.7.2. Cristallographic Experiments

Crystals of harpagoside derivatives were obtained by solvent exchange: the unpurified synthesized pentaacetylated harpagoside was dissolved until saturation in 1 mL of methanol in a small recipient located in a larger one containing ethanol. After 5 days, crystals formed thanks to the diffusion of the two solvents recovered. A needle shape crystal was selected, damped in a paraffin mixture, mounted on a nylon cryo-loop then centered on a Bruker-AXS-enraf-nonius KappaAPEXII goniometer equipped with a high brilliance micro-source. The data was collected with φ and ω scans, then integrated and corrected for the Lorentz and polarization effects using the EVAL14 software. Final cell parameters were obtained post-refining the whole data. The data was then reintegrated and corrected for absorption using the SADABS program and finally merged with the software XPREP. The structure was solved by direct methods and refined by full-matrix least square methods with, respectively, the SHELXT-2016 and SHELXL-2013 programs implemented in Olex2 software. C and O atoms were refined with anisotropic thermal parameters. H atoms were set geometrically, riding on the carrier atoms, with isotropic thermal parameters.

Crystal data for C_34_H_40_O_16_: monoclinic, space group P21. a (Å) = 16.681(3), b (Å) = 5.8089(12), c (Å) = 18.611(4), β (°) = 105.11(3), V (Å^3^) = 1741.1(7), Z=2, D (g.cm^−3^) = 1.344. λ (Å) = 0.71073. F(000) = 744. µ (mm^−1^) = 0.108. T (K) = 200. θ range (°) = 1.462–27.50. Measured, unique and used reflections: 28454, 7712 and 5136. [R(int) = 0.0682]. A total of 496 parameters. R(1)[I > 2σ (I)] = 8.84 %. WR(2) = 22.58%. G. O. F. = 1.024. CCDC 2226168 contains the supplementary crystallographic data for this paper. These data can be obtained free of charge via http://www.ccdc.cam.ac.uk/conts/retrieving.html accessed on 13 December 2022, (or from the CCDC, 12 Union Road, Cambridge CB2 1EZ, UK; Fax: +44-1223-336-033; E-mail: deposit@ccdc.cam.ac.uk).

### 2.8. Biological Activities

#### 2.8.1. Cells

The NHEK cell line were obtained from Bioalternatives (K341, K1087 and K1089).

HEK293T (Cat. # CRL-3216, RRID:CVCL_0063), MES-SA (Cat. # CRL-1976, RRID:CVCL_1404) and MES-SA/Dx5 (Cat. # CRL-1977, RRID:CVCL_2598) were purchased at American Type Culture Collection (ATCC)). The cell line HeLa GFP-LC3 was a kind gift from Dr. M. Faure (ENS-Lyon, France) [22,23,24].

Cells were maintained in DMEM (for HEK293T), McCoy’s 5A (for MES-SA and MES-SA/DX5) or RPMI-1640 (for HeLa GFP-LC3 cells) medium supplemented with 10% heat inactivated fetal bovine serum (Hyclone) and 1% penicillin/streptomycin (Sigma Aldrich) and grown in 5% CO2 at 37 °C in a humidified incubator. HeLa GFP-LC3 cells were cultured in presence of 0.5 mg/mL geneticin, and MES-SA/DX5 cells were grown with 500 nM doxorubicin.

#### 2.8.2. Tyrosinase Activity Assay

Extracts were added to a 100-µL final volume containing five units of tyrosinase from mushrooms (Sigma T3824-25KU) in 50 mM monobasic potassium phosphate buffer, pH 6.5 at 25 °C. After 5 min of incubation at room temperature, an initial absorbance at 477 nm was recorded and 110 µL of 12 mM L-DOPA (#CAYM13248-5, Cayman Chemical) in the same buffer were added. Then, an absorbance at 477 nm was recorded every 15 s for 15 min. Bioinactive and bioactive controls were based on wells with or without tyrosinase, respectively. For data analysis, absorbance values at reading time giving the highest Z’ factor [25] were considered, and the initial absorbance value was subtracted from the time-related absorbance value for each well. Results are expressed as percentages of tyrosinase inhibition relative to the bioinactive controls (with tyrosinase, i.e., 0% inhibition) and bioactive controls (no tyrosinase, i.e., 100% inhibition).

#### 2.8.3. Viability Assay

HEK293T, MES-SA and MES-SA/Dx5 cells were seeded at 2500 cells/well into 384-well plates (#781086 Greiner black plates). Twenty-four hours later, cells were treated with extracts, or with DMSO at the same final dilution as in the extracts. Bioinactive and bioactive controls were based on wells with or without cells, respectively. After 72 h of treatment, the PrestoBlue^®^ fluorescent cell viability reagent (ThermoFisher, Illkirch, France) was added to each well according to the manufacturer’s recommendations, and the fluorescent signal was quantified after a 30 min incubation. Results are expressed as a cytotoxicity percentage relative to the bioactive controls (diluted DMSO without cells, i.e., 100% cytotoxicity) and bioinactive controls (diluted DMSO with cells, i.e., 0% cytotoxicity).

#### 2.8.4. IL-8 Secretion Assay

HEK293T were seeded at 20,000 cells/well into 96-well plates (#655101 Greiner clear plates). Twenty-four hours later, cells were treated with extracts. Bioactive and bioinactive controls were based on cells treated or not treated by 10µg/mL TNFα, respectively, and with DMSO at the same final dilution as in the extracts. After 48 h of treatment, the secreted IL-8 in the cell supernatants was quantified using the Human IL-8/CXCL8 ELISA Kit (#RAB0319, Sigma-Aldrich) according to the manufacturer’s recommendations. Results are expressed as percentage of IL-8 induction relative to the bioinactive controls (no TNFα, i.e., 0% induction) and bioactive controls (10 µg/mL TNFα, i.e., 100% induction).

#### 2.8.5. NF-κB Nuclear Translocation Quantification

HEK293T cells were seeded at 10,000 cells/well into 0.001% Poly-L-Lysin precoated 96-well plates (#3536219 BD Falcon black clear bottom plates). Twenty-four hours later, the cells were treated with 0.5% DMSO or extracts for 2 h. The NF-κB pathway was activated with the addition of 10 ng/mL TNFα for 30 min (1.5 h after compounds or extracts addition). Bioactive and bioinactive controls were composed of cells treated with or without TNF-α, respectively. Cells were washed, fixed with 4% paraformaldehyde for 15 min and permeabilized with 0.5% Triton X-100 for 5 min. After 1 h saturation in PBS/5% horse serum, cells were immunostained with a rabbit anti-NF-κB p65 primary antibody (#sc-372, Santa Cruz, CA, USA) at 1/500 in PBS/5% horse serum for 1 h, washed three times with PBS/0.1% Tween-20 and immunostained with a goat anti-rabbit Cy3 secondary antibody (#111-165-144, Jackson ImmunoResearch Laboratories, West Grove, PA, USA) and 1 μg/mL Hoechst (#33342, Sigma-Aldrich) for DNA labelling, at 1/1000 in PBS/5% horse serum. Cells were finally washed three times in PBS/0.1% Tween-20 and conserved at 4 °C in PBS/50% glycerol until automated imaging. Eight images per well were acquired on an ArrayScanVTI (Thermo Scientific) with a Zeiss 10× (NA 0.4) LD Plan-Neofluor objective. The dichroic mirrors used for Hoechst and NF-κB were XF53-386/23 and XF53-572/15, respectively. Exposure times were set on control wells (with or without TNFα) to reach 25% and 40% of the fluorescence intensity of Hoechst and NF-κB staining, respectively. Quantification of the cytoplasm-to-nucleus translocation of NF-κB was made using the Molecular Translocation Bio-Application of Thermo Scientific HCS Studio v6.5.0. Briefly, individual nuclei are identified in the Hoechst channel based on the isodata thresholding method (border-touch nuclei are rejected) and used to create a Circ mask in the NF-κB channel. A Ring region is defined in the cytoplasm beyond the nuclear region. The average pixel intensity is measured within the Circ and Ring masks, leading to the CircRingAvgIntenRatio parameter. Results are expressed as a percentage of NF-κB nuclear translocation activity relative to the bioactive control (0.5% DMSO with TNFα).

#### 2.8.6. Autophagy Quantification

Autophagy quantification was performed as described in Jacomin et al. [26]. Briefly, HeLa GFP-LC3 cells were seeded on 96-well plate (#655090, Greiner) at 7500 cells per well; 24 h later, cells were treated with DMSO at 0.25% and rapamycin at 250 mM, as bioinactive and bioactive controls, respectively, or with the extracts at the indicated dilution, for 2 h. After fixation (4% paraformaldehyde), the DNA was stained with 1 µg/mL Hoechst. Image acquisitions were performed on an ArrayScan^VTI^ using a Zeiss 20× Plan-Neoflur air objective (10 fields per well). Images were automatically analyzed with SpotDetector Bio-Application of Thermo Scientific (Illkirch, France) HCS Studio v6.5.0, allowing us to count and extract the GFP-LC3 dots per cell parameter. Results are presented as a proportion of the cells containing less than three autophagosomes (ATG), between three and eight, and more than eight ATG.

## 3. Results and Discussion

### 3.1. Quantitative Chemical Composition of A. pyramidalis

#### 3.1.1. Preparation of the Extracts, Preliminary Studies and Primary Metabolites Quantification

The air-dried aerial parts of *A. pyramidalis* were ground into powder and then sequentially extracted with a triphasic solvent system based on a recent maceration procedure [18]. Three extracts of increasing polarity were thus obtained, namely apolar, intermediate and polar extracts, which, respectively, accounted for 4%, 14% and 34% of the dry plant matter. The overall yield of the extraction was therefore 52%.

First, to ensure the reliability of the quantitative studies, the residual quantities of water and solvents in these three crude extracts were determined. The dry matter was measured after the extracts were oven dried for 3 h at 110 °C: for the apolar, intermediate and polar extracts, the dry matter represented, respectively, 79.8 ± 0.2, 85.8 ± 0.8 and 75.5 ± 0.2% of the extracts. These results were then used to adjust the assays of quantification.

Before the specific analyses of each extract, some non-specific quantifications were simultaneously undertaken, i.e., the determination of ashes, the quantification of free amino acids, the assay of carbohydrates, the quantification of nitrogen compounds and of the lipid fraction. The results were gathered in Table 1.

The free amino acid quantification was assessed by the high-performance liquid chromatography (HPLC) technique, with a diode-array (DAD) and/or fluorescent detector (FLD). The 20 essential amino-acids and hydroxyproline were determined: they were all identified in the plant extracts, except lysine.

The nitrogen compounds quantity was evaluated by the Kjeldahl method [27,28]. Through this quantification, it is possible to approximate the protein percentage in the dry extract by subtracting the free amino acid percentage and by using a conversion factor of 6.25 (average ratio of nitrogen in proteins). Thus, the protein percentage was found, respectively, of 2.4% and 2.3% in the intermediate and polar extracts.

The carbohydrate quantification was achieved by external calibration using a mixture of rhamnose, arabinose, fructose, glucose and sucrose. The exact composition of each extract is given in Table 2.

In order to complete the above analysis on carbohydrates, a size-exclusion chromatography was operated on the polar extract and allowed the identification of polysaccharides (17%), even though the exact structure of these compounds was not ascertained.

The lipid fraction was determined after extraction of 500 mg of each crude extract with 5.0 g of heptane (Table 1). The results highlight that the apolar phase contains almost all of the apolar compounds of *A. pyramidalis*, together with other molecules. Subsequently, the extracts were derivatized (see experimental Section 3.4) and preferentially studied with gas chromatography (GC) using electron ionization mass spectrometry (EI-MS) and a flame ionization detector (FID) for the quantification of the lipidic content: 19 compounds listed in Table 3 were identified (fatty acids were detected through their methyl ester derivatives). From an experimental point of view, it is worth noting that the compounds identification was classically achieved thanks to the LC-MS-MS analyses before the subsequent quantification by the GC or HPLC methods.

#### 3.1.2. Secondary Metabolite Studies

Based on the phytochemical profiles of each extract (Figure 2), the quantification of the main identified secondary metabolites, namely organic acids, iridoids, phenolic and terpenic derivatives, was undertaken. Organic acids were found in the polar extract, whereas the three other families of metabolites were only detected in the intermediate extract.

The organic acid content of the polar extract was thus studied with HILIC chromatography using a time of flight sensor (ToF) and an evaporative light scattering detector (ELSD) through external calibration, thanks to a mixture of gluconic, glucuronic, tartric, oxalic, malic and *cis*-aconitic acids. The composition of the polar extract is presented in Table 4. Overall, this extract contains 8.9 ± 0.1% of organic acids.

The identification and quantification of the iridoid, phenolic and terpenic derivatives was achieved in the intermediate phase by chromatography using DAD, ELSD and HRMS detectors. MS/MS fragmentation was also used to define molecular formulas or even structures of these families of compounds (Table 5). Regarding the phenolic derivatives, the UV spectra of each compound were compared to an internal data bank to identify three of them. Finally, the purification of five of these compounds (**1d**, **4d**, **1-3e**, see below paragraph 2.2) allowed to unambiguously confirm their structures after NMR analyses. Overall, this extract contained 16.3 ± 0.1 and 12.3 ± 0.1% of iridoids and phenolic acids (or phenylethanoid glycosides), respectively. The terpenic derivative content was not determined but fragment ions of compounds **1f** and **2f** could match with clerodane diterpenes. In particular, compound **1f** (C_27_H_38_O_9_) could be neoajugapyrin A, which was previously identified in an extract of *A. pyramidalis* [16].

#### 3.1.3. Global Composition of *A. pyramidalis*

In the end, more than 95% of the extracted mass has been characterized. More specifically, 98% of the apolar extract has been determined (among which 93% by FID). It is mostly composed of fatty acids, alkanes, stigmasterol derivatives, minerals and carbohydrates. The intermediate extract has been described at 92%, and mainly includes carbohydrates, iridoids and phenolic compounds, whereas 97% of the polar extract has been labelled. The latter predominantly contains carbohydrates, minerals and organic acids. Overall, 64 molecules, including essential amino-acids, have been identified. Previous quantifications were finally gathered in Appendix A (see Appendix A) taking into account the proportion of each extract (in view of the dry matter).

The unknown fraction in the intermediate extract could be iridoids, terpenic derivatives or other compounds that have not been determined, mainly because they were not detected or because their quantity was too low (detection threshold issues).

### 3.2. Isolation, Purification and Structural Characterization of Iridoids and Polyphenol Derivatives

Simultaneously to these quantification studies, the intermediate extracts, which contain interesting secondary metabolites, were sequentially fractionated. The goal was twofold: to ascertain the structure of the identified major compounds after the purification and NMR analyses, and to evaluate the biological properties of some enriched extracts or even pure compounds.

Purification was thus processed on this extract and five compounds were isolated after fractionation by normal phase and reverse phase chromatography. Then, the identification was achieved through HRMS mass spectrometry (Table 5) and NMR analyses (^1^H, ^13^C, HSQC, COSY and HMBC, see spectra in Appendix A) while referring to the bibliographical data describing these compounds or analogues. Three polyphenols, namely echinacoside, verbascoside and teupolioside, and two iridoids (8-*O*-acetylharpagide and harpagide) were ultimately purified and characterized (Figure 3).

To go deeper in this characterization, efforts of crystallization of the main purified iridoid, 8-*O*-acetylharpagide, have been made. Unfortunately, all attempts failed: some crystals formed but their size or their quality was not satisfactory enough to resolve their structure. Further tests were not possible because the remaining quantity was too low. However, since no crystal structure of the harpagide derivatives was found in the literature, we decided to try to crystallize a commercially available member of this family: harpagoside. To promote crystallization, the derivatization of harpagoside was undertaken (Figure 1). An acetylation was carried out in presence of eight equivalents of acetic anhydride in pyridine. After 4 h at 70 °C, the reaction was not completed and the reaction mixture was then stirred for 7 days at room temperature. Even if the HPLC analyses showed that several products were still present (see Appendix A), the reaction mixture was treated since a major compound could be identified. Even before the purification of this major acetyl derivative of harpagoside, crystallization tests were made and successfully provided crystals that gave access to the crystal structure by RX of a pentaacetylated harpagoside derivative (Figure 4).

### 3.3. Biological Studies

#### 3.3.1. Biological Evaluation of the Three Extracts

Plant extracts are privileged sources of compounds with benefits for pharmaceutical or cosmetics applications [29,30,31,32]. The bioactivity of apolar, intermediate and polar extracts were thus evaluated by monitoring their potential to inhibit tyrosinase in vitro as well as their impact on cultured human cells regarding cell viability, inflammatory response or autophagy induction.

Tyrosinase is a regulator of skin pigmentation in melanocytes and a recognized target for altering skin color in various cosmetics applications [33]. In order to detect a potential impact on tyrosinase activity, extracts were tested on purified tyrosinase in an enzyme-based assay at three dilutions, ranging from 1/200 to 1/2000. None of the extracts showed a significant effect on tyrosinase activity, at any dilution (Table 6).

Cosmetic or pharmaceutical use of plant extracts and compounds requires either weak or no toxicity towards normal human cells or tissues, versus, in the case of applications in cancer field, a high toxicity towards cancerous cells. To evaluate a potential impact on cell viability, the extracts were tested on three different human cell lines: the renal embryonic HEK293T cell line, the uterine sarcoma MES-SA cell line and the MES-SA-derived MES-SA/Dx5 cell line, which correspond to non-cancerous, cancerous and multi-drug-resistant models, respectively (see methods). Cells were treated with the extracts at three dilutions ranging from 1/200 to 1/2000, for 72 h before quantification of the cell viability. At the lowest dilution of 1/200, all extracts displayed from moderate (i.e., in the range of 30%) to high toxicity (from 50% to 90%) on all cell lines tested. The apolar extract was still partially toxic at the 1/667 concentration on both the HEK293T and MES-SA Dx5 cells. Strikingly, at the lowest dilution, the polar extract was mainly toxic on cancerous cells (65.1% and 45.3% of toxicity on MES-SA and MES-SA/Dx5, respectively) compared to non-cancerous HEK293T cells (21.8%), an observation that may deserve further investigation (Table 6). At higher dilutions, the three extracts, except the apolar extract at 1/667 as stated above, showed no cytotoxicity towards both non-cancerous and cancerous cell lines. This result endorsed further biological evaluations of these extracts on living cells at non-toxic doses (in the range of 1/667 to 1/2000).

The HEK293T cells are immune competent and secrete basal levels of IL-8 even in the absence of a specific inducer. The pro-inflammatory potential of the extracts was first evaluated on these cells through the quantification of basal IL-8 secretion. Cells were treated with the extracts at dilutions ranging from 1/200 to 1/2000, for 48 h before the quantification of secreted IL-8. None of the extracts induced any enhanced basal secretion of IL-8 suggesting no spontaneous pro-inflammatory effect (Table 7). The polar extract showed a mild inhibition of basal IL-8 secretion at 1/200 and 1/2000, but with no clear dose-response effect, suggesting that it is not relevant.

In eucaryotes, inflammation is mainly triggered by the activation of Nuclear Factor kappaB (NF-κB), a family of transcription factors activated downstream of various plasma membrane receptors, such as the Tumor Necrosis Factor alpha (TNFα) receptor (TNF-R). NF-κB is mostly present and sequestered in the cytoplasmic compartment where it is inactive while its activation depends on the proteasomal degradation of its inhibitor I-κB, allowing its subsequent translocation to the nuclear compartment where it regulates the expression of a huge amount of genes encoding pro- or anti-inflammatory cytokines and survival factors [34].

In order to further evaluate their pro- or anti-inflammatory potential, we tested the three extracts at non-toxic doses from 1/667 to 1/6667 dilution on the nuclear translocation of NF-κB/p65 upon stimulation with TNFα using automated microscopy and images quantification (Table 7, see methods). While NF-κB/p65 is fully sequestered in the cytoplasm of control cells (“0.5% DMSO -TNFα”, Figure 5A), TNFα stimulation resulted in strong staining in the nuclear compartment in addition to the cytoplasm (“0.5% DMSO + TNFα”, Figure 5A). The intermediate and polar extracts had no effect on the nuclear translocation of NF-κB as observed after 30 min of stimulation with TNFα, whatever the dilution tested (Table 7 and Figure 5A). However, the apolar extract increased the nuclear staining (to 147% that of the untreated control cells) at the 1/2000 dilution (Table 7). These results indicate that the two most polar extracts would have neither anti- nor pro- inflammatory activity in the HEK293T cells, while the apolar extract may enhance or potentiate the pro-inflammatory response induced by the TNFα.

Autophagy is a cellular “self-digestion” process that is activated in eukaryotic cells in response to nutrient deprivation or other cellular stress and pathological conditions, such as a microbial infection or the accumulation of toxic aggregates in neurodegenerative disorders, etc. [35]. It is characterized by the formation of autophagosomes (ATG) capturing organites, protein aggregates or microbes that will fuse to lysosomes, thus triggering their digestion in this acidic compartment [36]. We evaluated the amount of autophagosomes in HeLa GFP-LC3 stably transfected cells by automated microscopy and image quantification using the expression of a GFP-LC3 fluorescent protein that is specifically recruited in these intracellular structures (Figure 5B, see methods) [26]. We observed that when treated during 2 h with the diluted extracts, HeLa GFP-LC3 cells exhibited the same autophagosomes distribution profile as the control (i.e., non-treated cells in 0.25% DMSO): 50–60% of cells contained less than three autophagosomes, 30–40% between three and eight, and a very low proportion of cells displayed a higher number of autophagosomes (Figure 6). This result indicates that cells treated with each of the extracts have a normal basal level of autophagosomes, suggesting that they did not induce major stress when applied to living cells.

#### 3.3.2. Epidermal Renewal Evaluation

Besides the above-mentioned biological assessment, a targeted evaluation for cosmetic use was led to investigate the activity of some extracts and pure compounds on the expression of different genes. Six samples were thus tested: the three different extracts (apolar, intermediate and polar), a fraction containing the four iridoids, a fraction containing the five polyphenols identified in the intermediate extract, and purified 8-*O*-acetylharpagide. These samples were evaluated on normal human epidermal keratinocytes (NHEK) from three different donors by the expression analysis (mRNA) of 44 markers selected by biologists at CHANEL Parfums Beauté (see Appendix A for complete results table—Appendix A).

Firstly, the cytotoxicity was determined using the CHANEL Parfums Beauté internal protocols for each sample on this particular cell-line—as cytotoxicity may depend on the cell line used—to select the maximum non-toxic concentration usable for the subsequent assays. The apolar extract was cytotoxic at concentrations greater than 0.001%, the intermediate extract at 0.03% and the polar extract at 0.19%. The iridoid fraction was cytotoxic at 0.027%, the polyphenols fraction at 0.003% and the purified 8-*O*-acetylharpagide at 0.007%.

Concerning the apolar extract, due to its high cytotoxicity, it was tested at a low concentration of 0.001% that did not allow the observation of any impact on the genes.

For the polar extract, K10 gene expression was stimulated in one donor and tended to be stimulated in the second donor (factor 1.61). In addition, the DSC1 gene was stimulated in both donors. These two genes are two biological markers involved in the process of differentiation of the epidermis. In addition, the DSC1 gene, which encodes a protein involved in the formation of cell–cell junctions called desmosomes and is important in the epidermal differentiation process, was stimulated in both donors. These results suggest that the extract could stimulate the epidermal differentiation stages. From a cosmetic point of view, this type of extract would allow a targeted action on strengthening the barrier function of the skin and would also participate in the maintenance of skin hydration.

The intermediate extract had little effect at the concentration tested. Only the TGM1 target, an enzyme involved in the formation of the corneal envelope of keratinocytes during the final stage of differentiation, was similarly modulated in both donors (decreased expression). This extract had also a tendency to decrease the expression of the HSPB1 gene, corresponding to a protein involved in the stress response and preferably expressed in differentiated epidermis cells. Since these biological targets are reduced in keratinocytes, these results suggest that the differentiation process is not favored and suggest pro-proliferative activity.

The polyphenol fraction showed little effect at the tested concentration. However, a decrease in the expression of TGM1 and HSPB1 is visible in both donors. This signature was similar to the intermediate extract. In addition, the expression of CLDN1 was decreased in one donor and tended to be decreased in the second donor (0.55). CLDN1 is a protein involved in the formation of cell–cell junctions, called tight junctions, and is important in the process of epidermis terminal differentiation. Finally, the expression of NOTCH1 was decreased in one donor and tended to be decreased in the second donor (0.57). NOTCH1 is involved in the regulation of epidermal homeostasis (the balance between proliferation and cell differentiation). NOTCH1 slows down cell proliferation and stimulates the early stages of differentiation in the epidermis. As NOTCH1 is diminished, the proliferation process may be encouraged. These results reinforce the hypothesis of a pro-proliferative extract with a decrease in cell differentiation.

The iridoids fraction, at the tested concentration, modulated 11 genes similarly for the two donors. Among these genes, it was possible to find a decrease in the expressions of TGM1, CLDN1, DSC1, LGALS7, HSPB1 and NOTCH1 that are associated with the cell differentiation processes. These observations suggest a strongly pro-proliferative impact on keratinocytes. In addition, this extract stimulated the expression of NQO1 in both donors and HMOX1 in one donor, two genes that are involved in the oxidative stress response and cellular defense system. These results suggest that this extract could stimulate the proliferation of keratinocytes as well as the response to oxidative stress. From a cosmetic point of view, this type of compound would allow a targeted action on the renewal of the epidermis and participate in the skin defense in an anti-aging approach.

The purified 8-*O*-acetylharpagide had little effect at the tested concentration. Nevertheless, it was possible to observe a similar trend of activity on the various active genes to the one of the iridoid fraction.

### 3.4. Complementary Exploration: Intra- and Inter-Species Studies

#### 3.4.1. Intra-Species Studies

Two intra-species studies were conducted on *A. pyramidalis*: the first one concerns the interannual variability, while the second aims at comparing the chemical composition of the different plant parts.

For the interannual variability study, samples of *A. pyramidalis* were collected at the same place and a similar plant development stage for three consecutive years. The phytochemical profiles of simple ethanolic extracts were then compared by HPLC-DAD and ELSD (see Appendix A for chromatograms—Appendix A). Whatever the year of harvest, the phytochemical profiles are very similar with only slight variations of approximately 7–7.5 min on the HPLC-DAD chromatograms and at 11.5 min on the HPLC-ELSD chromatograms: this underlines some stability in the chemical composition of the plants, a relevant factor when exploiting the biological properties of the plants.

In the second study, the chemical contents of the roots, flowers and the remaining aerial parts of the plants were compared. The phytochemical profiles of the ethanolic extraction of each part of the plants highlighted similarities for the flowers and the aerial parts (without flowers), except in the polar region: the content in polar compounds is higher for flowers, usually richer in carbohydrate derivatives (Figure 7). In the roots, their phytochemical profile is significantly different, mainly in the zone of the secondary metabolites between 5.2 and 6.5 min.

#### 3.4.2. Inter-Species Study

France comprises only five species of the genus *Ajuga* L., namely *A. reptans* L., *A. genevensis* L., *A. pyramidalis* L., *A. chamaepitys* (L.) Schreb. and *A. iva* (L.) Schreb. In this study, only the first four species were investigated. After being collected and extracted with ethanol, the HPLC-DAD and ELSD chromatograms were recorded for each plant extract (Figure 8). Even if the same general shape can be observed, the different families of secondary metabolites are not recovered in the same proportions. Between 5.5 and 6.5 min, which includes the phenolic acid region, *A. pyramidalis* chromatogram presents two main peaks (echinacoside and verbascoside) that are not found, or are found to a lesser extent, in other species, in particular in *A. chamaepitys*. Nevertheless, the latter counterbalances its phenolic acid poverty by an increased amount of iridoids (between 2.2 and 5.2 min), especially harpagide. Finally, some terpenic derivatives can only be found in *A. pyramidalis* (at approximately 6.6 min).

## 4. Conclusions

In summary, all these assays and quantifications allowed us to quantify 95% of the mass extract from *A. pyramidalis*, which can mainly be distributed in 64 identified compounds, including five purified compounds. Among them, three compounds never isolated from *A. pyramidalis* but known in other plants of this family were characterized, namely echinacoside, verbascoside and teupoloside. The first RX structure of a harpagide derivative, the pentaacetylated harpagoside (free hydroxyl at position 4a), was also obtained during this study. Furthermore, the different fractions of the plant were investigated on purified tyrosinase as well as on different kinds of human cultured cells regarding their potential activity on major cellular processes, i.e., cell viability, inflammation, stress response or epidermal renewal, and they demonstrated some biological activities that could be further harnessed for cosmetics use.

## Data Availability

All experiments are described directly in this article and/or the original procedures are referenced in the text; except the biological procedures for the epidermal renewal part. These evaluations were performed by Chanel PB following well established internal procedures.

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
