# Peer review of "Investigation of Chemical Composition and Biological Activities of Ajuga pyramidalis—Isolation of Iridoids and Phenylethanoid Glycosides"

_metabolites, 2023, doi:10.3390/metabo13010128_

Round 1

Reviewer 1 Report

The manuscript " Investigation of Chemical Composition and Biological Activities of Ajuga pyramidalis – Isolation of Iridoids and Phenylethanoid Glycosides" by Gori et al., describes phytochemical pattern and biological properties of Ajuga pyramidalis. Overall, the team has performed extensive analysis, and the manuscript is well organized. I have a few suggestions. Please find my comments below. 

Line #47 Elaborating more about medicinal use will be helpful. Authors should reduce the following paragraph about the general information and geographical distribution. Authors can even consider eliminating the section (Line #49 - Line #57)

Figure #3 Few words are not readable. There is a scope for improvement in the presentation. Also, the authors should briefly describe the figure; for example, is there a significance of different color coding?

Fig. 6 Microscopy images should be colored. Also, is the staining specific for NF-kB p65 or p65-p50 heterodimer? 

It will be nice to have immunoblot analysis of p65 nuclear localization in cytoplasm and nuclear extracts/fractions.

Method 3.8.5 Authors should add the catalog numbers of all the antibodies used.

3.8.6. Apart from the Jacomin et al., reference, please describe the method briefly in 2-3 lines. 

Author Response

The manuscript " Investigation of Chemical Composition and Biological Activities of Ajuga pyramidalis – Isolation of Iridoids and Phenylethanoid Glycosides" by Gori et al., describes phytochemical pattern and biological properties of Ajuga pyramidalis. Overall, the team has performed extensive analysis, and the manuscript is well organized. I have a few suggestions. Please find my comments below.

Answer: We thank the reviewer for his kind comment.

Line #47 Elaborating more about medicinal use will be helpful. Authors should reduce the following paragraph about the general information and geographical distribution. Authors can even consider eliminating the section (Line #49 - Line #57)

Answer: The sentence “In particular, some Ajuga species have been traditionally used in ethnomedicinal practices for anti-inflammatory activity, expectorant effect, protective action against heart diseases, stomach pain or for the treatment of jaundice, asthma or even cancer” was added at the end of the first paragraph to exemplify some traditional uses of Ajuga species. Moreover, the section (Line #49 - Line #57) was largely shortened.

Figure #3 Few words are not readable. There is a scope for improvement in the presentation. Also, the authors should briefly describe the figure; for example, is there a significance of different color coding?

Answer: This figure has been transferred in supporting information: it has been also changed for a higher quality image and enlarged. Color information was added in the title of the figure.

Fig. 6 Microscopy images should be colored. Also, is the staining specific for NF-kB p65 or p65-p50 heterodimer? 

It will be nice to have immunoblot analysis of p65 nuclear localization in cytoplasm and nuclear extracts/fractions.

Answer: We now provided a colored figure. We indeed understand that it can help the reader to distinguish different kinds of staining. However, we would like to point out that microscopy images were presented in white and black as the best image quality standard according to the acquisition in grey mode, thus providing a better resolution compared to colored figures. Colored figures are rather used when merge staining are presented which was not our choice here.

We have chosen the analysis of NF-κB nuclear translocation as the most informative methodology to monitor NF-κB paths activation while quantifications by western blots are sometimes very tricky. We have a strong experience in monitoring NF-κB activation and in our hands, HCS technology revealed itself as the most relevant methods.

The staining for NF-κB is specific of p65. This information has been added in Table 7, Figure 5 and in the results (Lines #304-305). The reference of the anti-NF-κB p65 antibody has been added in Method 3.8.5 (Line #630).

Method 3.8.5 Authors should add the catalog numbers of all the antibodies used.

Answer: Catalog numbers of anti-NF-κB p65 antibody and goat anti-rabbit Cy3 secondary antibody have been added (Lines #630 and #632 respectively).

3.8.6. Apart from the Jacomin et al., reference, please describe the method briefly in 2-3 lines. 

Answer: The method used for autophagy quantification (p3.8.6) has been implemented.

“Briefly, HeLa GFP-LC3 cells were seeded on 96-well plate (#655090, Greiner) at 7.500 cells per well; 24 h later, cells were treated with DMSO at 0.25% and rapamycin at 250 mM, as bioinactive and bioactive controls respectively, or with the extracts at the indicated dilution, for 2 h. After fixation (4% paraformaldehyde), DNA was stained with 1 μg/mL Hoechst. Image acquisitions were done on an ArrayScanVTI using a Zeiss 20x Plan-Neofluor air objective (10 fields per well). Images were automatically analyzed with the SpotDetector Bio-Application of Thermo Scientific HCS Studio v6.5.0, allowing to count and extract the GFP-LC3 dots per cell parameter.”

Reviewer 2 Report

Paper is well written, presenting interesting spectral  data of iridoids isolated from. A. pyramidalis.

Some minor remarks:

1. Correct the name of 8-O-acetylharpagide, on Fig. 1; 

2. Lipidic content and relative quantification were quantified on 100% polydimethylsiloxane (Agilent VF-1MS) column? If yes, the separation of some fatty acid on this column is not so good, as on polar. 

3. Precise, if you analyzed fetty acids and terpenes (such as amyrin) or corresponding methyl ester (see point 3.4);

4. According to NIST20, KI for a-amyrin is around 2873 on such type of column. Please clarify;

5. According to NIST20, KI for stigmasterol  is around 3170 on such type of column (in Table 3351). Please clarify;

6. If possible, please add carbon numeration for harpagide for better understanding compounds structure.

7. In p. 3.4. precise, what extract was dissolved in MeOH?  

Reviewer 3 Report

I read the manuscript “Investigation of Chemical Composition and Biological Activities of Ajuga pyramidalis – Isolation of Iridoids and Phenylethanoid Glycosides” carefully and realized that it needs a major revision. The title of this paper sounds good, However, some ambiguous points within this submission should be addressed, modified, or clarified in its revised form.

1.     English: I encourage authors to have your manuscript checked and corrected by a fluent or native speaker of English

2.     There are too many figures. The figures can be combined together or moved to support information (SI).

3.     In some Figures/Tables', the legends should contain a basic description of n, and the statistical test applied. Are the values presented as mean +- SD or SEM?  Please provide more information in the legends for the Figures and Tables.

4.     Keywords are too many decrease them as possible.

5.     From lines 33 to 44 kindly delete, and I think no need to describe the Lamiaceae family

6.     The authors have to use LC-MS-MS instead of GC and HPLC methods which are not intended for your work identification purposes. 

Round 2

Reviewer 3 Report

The authors did all the required corrections and modifications and I have no more comments

Author Response

Thank you for the revision.

Best regards